# Encounter with Bullying in Sport and Its Consequences for Youth: Amateur Athletes’ Approach

**DOI:** 10.3390/ijerph16234685

**Published:** 2019-11-25

**Authors:** Jolita Vveinhardt, Vilija Bite Fominiene, Regina Andriukaitiene

**Affiliations:** 1Institute of Sport Science and Innovations, Lithuanian Sports University, Sporto St. 6, LT-44221 Kaunas, Lithuania; jolita.vveinhardt@gmail.com; 2Department of Sport and Tourism Management, Lithuanian Sports University, Sporto St. 6, LT-44221 Kaunas, Lithuania; regina.andriukaitiene@lsu.lt

**Keywords:** athletes’ behavior, organized sport, antisocial behavior, bullying, victims, bully, Lithuania

## Abstract

In recent years, the problem of bullying, existing in sport and arising in athletes’ relationships, is increasingly emphasized. The aim of this research was to reveal the specificity of bullying in athletes’ interrelationships by elaborating on causes of its emergence, nature of actions, and its consequences. To achieve the research aim, a qualitative research paradigm was chosen. The theoretical part of the research was prepared by applying the methods of scientific literature analysis and analogy. The empirical study involved seven organized sports athletes representing individual, duel, and team sports branches, belonging to the young adult age category. The survey was conducted using the semi-structured interview method. Data were analyzed employing the conceptual content analysis. Emic and etic perspectives were used for data processing. Research results revealed that the specificity of manifestation of bullying in sport unfolded through three generalized categories: intolerable perception of behavior, nature of bullying, and bipolarity of consequences. Every category was detailed by sub-categories, highlighting the nature, causes, and consequences of bullying accepted by athletes in the contexts of their emotional state and career. We found that the factors falling into these categories were interrelated and supplemented each other; therefore, they should be analyzed in a complex way, as bullying is determined not by some single factor but by the totality of them, functioning as a kind of well-established flawed tradition supported by the cultures of the sports organization and the sport.

## 1. Introduction

Sport has undoubtedly become an integral part of modern society. Over the past decade, the field of sport-for-development, associated with the importance of using sport and in particular the importance of sport for personal and social development outcomes, started to receive considerable attention [1]. This is also found in research conducted in these areas. The studies name sport as an effective mechanism to promote positive youth development [2], and as a tool contributing to the creation of society’s well-being by enabling members of the society to become healthier and happier [3]. However, sport, as an integral part of society, also encounters a set of the same problems that are faced by society. One of them is bullying, the complex problem of social relationships, deeply rooted in our society.

Bullying is a unique but complex and heterogeneous phenomenon, which “directly affects hundreds of millions of people each year” [4] (p. 327). Persons who have directly encountered this phenomenon, that is, persons who have found themselves in the role of the victim of a bully report a negative impact on their physical, emotional/psychological, and socio-economical areas of functioning. However, the extent of the prevalence of bullying, the causes of its appearance, and the actions and consequences of bullying cannot be accurately identified because they depend on both the context or cultural differences and differences in measurement strategies [5]. For these reasons, particular attention is paid to the greater need for agreement in defining and analyzing bullying behavior [6,7], which will become a precondition for managing this negative phenomenon.

However, it is not easy to define this phenomenon accurately, as due to constant change and development of the society, it cannot remain static [8]. Often the term “bullying” is interchangeably used with the term “aggression” or referred to as intentional aggressive behavior, but it is acknowledged that such behaviour without a power imbalance cannot be understood as bullying behavior [9]. The conception that is best known and most widely used in scientific literature analyzing various manifestations of bullying is the one of Dan Olweus, originally proposed as early as in 1993, defining school bullying. Alongside basic characteristics—intentional aggression and power imbalance between the victim and the bully—it also emphasizes the repetition of the aggressive behavior. However, the phenomenon of bullying in human relationships and behavior is both observed in other contexts and manifests itself in interpersonal behavior of persons of various ages [10]. Volk et al. [4], in discussing this phenomenon, provided a definition that, in the authors’ opinion, can be perfectly used for defining bullying not only among adolescents but also among older and younger populations, being applicable to a wide range of domains. It is a concept that highlights three key attributes of bullying: aggressive goal-directed behavior, power imbalance, and victim harm. The concept of bullying should also be expanded through a sociocultural framework. This means that bullying is also determined by well-established interpersonal relationships within a particular social context [8]. An appropriate methodology in the studies on bullying behavior, which depend on the assumption of the researcher and on the nature of the phenomena under investigation, is also considered an important aspect of assessing the phenomenon of bullying. However, as early as 2012, Hong and Espelage stated that most studies analyzing bullying traditionally employed quantitative methodology, which also contributes to incomplete understanding/explanation of the phenomenon [7].

The topic of bullying in sport has also been increasingly analyzed in the past decade [11,12,13,14,15,16,17,18,19,20], which is to be related to the increased interest in participants’ behavior in sport and raised moral and ethical questions. However, the existing lack of research in the sport context, the use of various methodological approaches and instruments, and the incompletely purified treatment of the phenomenon [21] do not allow for the formation of a complete picture of both the extent of the problem or manifestation of the phenomenon and of consequences experienced by persons involved in it, as well as the ability to propose effective prevention and intervention measures. The problem is also exacerbated by the right granted to sport “to keep its own house in order” [22] (p. 17), which often leads to the situation that interpersonal relationship problems related to unethical or antisocial behavior are insufficiently emphasized and highlighted in empirical research.

It should be maintained that such an approach may be partly related to much lower prevalence of bullying in sport, comparing it with other contexts, especially school bullying, stated in research. Research results show that 10–15% of athletes in large part identify themselves as victims of peer bullying. An even smaller share of athletes (8–11%) attribute the role of the bully to themselves [11,15,16,18]. However, there are single studies showing a significantly higher prevalence of bullying in sport, which reveal that more than 20% of athletes’ state having experienced bullying [23]. This, according to a series of studies on the consequences of bullying, means that there is a high probability that these persons will eventually encounter depression, anxiety, suicidal behavior or ideation, poor mental and general health, and other problems [6,24].

It is evident that such a situation shows the need for effective intervention and prevention measures. Unfortunately, the latter are often not implemented in a qualitative manner in organizations operating in Lithuania, including sports organizations. This can be due to both imperfect legal acts or non-incorporation of existing ones into the organizational internal regulation legal acts [25] and the inability to recognize bullying or to know its specificity in a specific context, as well as it still being a relatively persistent situation when a wide range of bullying prevention activities are usually applied locally without any evidence-based research [26].

The imparted ideas enabled the following problem questions of the research to be raised: How is the phenomenon of bullying in interpersonal relationships in sport treated by athletes, that is, what can be the causes of the emergence of bullying? What is the specificity of manifestation of bullying in sport? What consequences of bullying are experienced and seen by athletes?

The research aim was to reveal the specificity of bullying in athletes’ interpersonal relationships, detailing the causes of its emergence, actions, and consequences.

## 2. Materials and Methods

The object of the research was bullying and its consequences in sport. Seeking to comprehensively analyze the research object and to implement the set research aim, the qualitative research paradigm was chosen, which enabled us to reconstruct subjective experience and, disclosing the context, to describe the process in which the concrete practice takes place. The study was grounded on the constructivist-interpretive paradigm [27], by which reality is conceived as a human construct formed by the participant’s cultural and personal life and does not exist without it. That is, the study served as a means to understand the world in which the subjects live and/or act, maintaining diversity and multiplicity of meanings rather than narrowing the phenomenon to several categories or ideas. The questions asked were broad, general, and open, so that enough space was left to reveal the meanings given by the investigated persons to phenomena analyzed. The researcher occupied the listener’s position, that is, the researcher carefully listened to what people were saying, the stories they were telling, and how they were acting in their life environment, focusing on specific things and contexts in order to understand people’s historical and cultural approaches. Attention was paid to how human experiences shape personal interpretations of meanings [27].

The sample was obtained using two different types of non-probability sampling techniques: purposive/judgmental and snowball [28]. Using the purposive/judgmental technique, the sample was formed after the researchers conducted field investigation in order to ensure that certain types of individuals matched certain features. In this case, the researcher had met athletes who had been identified as being bullied. The face-to-face meetings took place outside the sports club environment. The aim of the study was explained to them and the contact details of the researcher were given to them. Athletes who voluntarily expressed their willingness to reveal their experiences and attitudes were included in the study. A snowball technique was also used to find participants for the research. In this case, the subjects as persons who had experience and information on the topic of the conducted research were selected using the recommendations of other research participants. In this case, the athletes were contacted by telephone, e-mail, or face-to-face.

The main requirements for participants’ selection were as follows: encounter with the phenomenon of bullying in sport (the victim of bullying in the opinion of other persons); correspondence to the age group of 18–29 years, defined in Lithuania as “youth”; participation in organized sports for at least 5 years. In total, seven informants were recruited to the study and each of them received a code (e.g., 1S–7S) in the transcribed texts (Table 1).

Research data were collected using semi-structured face-to-face interviews with open-ended questions. The interview guides for athletes were created following guidelines proposed by H.J. Rubin and I.S. Rubin [29] and had introductory, main, and summary questions. Introductory questions were intended for making contact with the subject and collecting socio-demographic information. The main questions consisted of three parts focused on socially intolerable behavior (for example, “Remember the situations and talk about intolerable behavior in communication or relationships, manifesting itself in the team. By what actions and in what situations does intolerable behavior manifest itself?”), on the nature of bullying (for example, “What kind of bullying have you yourself experienced/are you yourself experiencing while doing sports? What kind of bullying related to other athletes have you yourself experienced/seen/are you yourself experiencing/can you see?), and on the consequences of bullying (for example, “What consequences could you name, what are they like personally for you as the athlete who has experienced bullying?”).

Interviews took place at selected places between January and February of 2019; their average duration was 32 minutes. Dictaphone was used to record all interviews. Interview records constituted the research data, which were transcribed verbatim by one of the authors of this article. After the transcription of the text (the total volume of the transcribed text in the Lithuanian language was 21,934 words), the electronic media of interview records were destroyed and the text was sent to informants for information, supplementation, and confirmation of correctness of the text.

Data were analyzed employing the conceptual content analysis [30]. After the preparation phase, in which units of the analysis were selected, the data was analyzed using an inductive approach. The main steps of this analysis consisted of open coding, coding sheets, grouping, categorization, abstraction, and reporting the analyzing process and results [31]. The categories distinguished in the coding process were based on emic and etic perspectives [32]. In order to ensure reliability of conclusions of the analysis, researchers’ interpretations were based on research participants’ “voice”, providing extracts from research interviews in the text. In order to ensure trustworthiness and credibility of the study and to reduce the risk of bias, all researchers were involved in the analytic process, purified topics were thoroughly discussed, and the interpretations provided were based on exhaustive quotations. Additionally, thorough interview guides prepared before starting the study ensured that the same questions would be given to investigated persons, this way avoiding interviewer bias.

The study was ethically approved by the ethics committee of the Lithuanian Sports University. The study was conducted in accordance with the principles of anonymity, confidentiality/participant safety, voluntariness, and authenticity of research data [33,34], that is, research participants were assured that the data collected during data-collection would not reveal their identity—name, surname, team name, and other information allowing their identification [33]. Research participants were also thoroughly acquainted with the research aim, the course and content of the research, and were informed about the use of research data and the publicizing of the obtained results. Study participants took part in the study voluntarily and were given the opportunity to withdraw from the study at any time [34]. The collected research data were not modified corrected and were accepted as valuable data that could affect any outcome of the study that was not predicted in advance.

## 3. Results

Analyzing research participants’ statements, it should be noted that answering initial questions, whether they have experienced or observed intolerable behavior in interrelationships, athletes gave negative answers, stating that relationships in the team or among persons training together were “friendly”. It was only in the course of interviews that the provided stories revealed experienced and observed intolerable behavior leading to bullying, with its nature and consequences showing up.

The first distinguished category was intolerable perception of behavior in interrelationships between athletes (see Table 2).

Aggression among athletes can be expressed using both psychological pressure and physical force. Psychological type of aggression was revealed in the narratives of representatives of three sports branches, highlighting several interrelated aspects. Answers of representatives of team branches 2S (football) and 3S (basketball) indicated unfair competition, when seeking to gain advantage, in attempting to degrade the competitor in the same team. In the narrative of 2S, one can discern certain subjectively perceived inferiority of the opponent, which was offset by degrading competitors. The explanation of 3S shows the nature of actions in more detail—formation of the negative opinion about team members and degrading their reputation by spreading rumors. The picture of the situation is also supplemented by the clarification provided by 5S (athletics) that sneers are used in direct contact, that is, seeking to cause pain openly. At the same time, it can be noted that such relationships have signs of being permanent in nature, leaving the only choice for the victim being to accept it (for example, “... if you have that thick skin, you don’t pay attention ...”). No broad generalizations can be made on the basis of these cases, but further research may verify the assumption that team sports may seek to employ group pressure, whereas in individual sports such opportunities remain more limited.

Physical aggression became evident in team sport branches (1S, rugby and 3S, basketball). Although 1S only reflected on experience observed from outside, the narrative of 3S shows subjective experience, which, on the one hand, reveals that aggression is used against the physically weaker person, and on the other, sounds like an excuse: “... smaller all the time <...> I couldn’t oppose”, which implicitly suggests the perception that opposition to aggression is only possible if you have “sufficient” physical force. The fact that physical aggression among team members may be frequent was confirmed by the reference to the presence of a person in the team who was especially responsible for “curbing” the team. In addition, it can be envisaged that the team is aware that interpersonal aggression occurs outside the public space (“in the changing room”), which cannot be directly controlled by team management; therefore, the team captain of standing is used for its suppression in this space. In this context, it is significant how athletes themselves assess cases of aggression (reaction to the use of physical force). Informant 3S, who experienced aggression, perceived this as “character hardening”. Other informants representing team sports treated the use of physical force in interpersonal relationships as a norm (1S “... we are not dancing ballet ...”) or tended to understate such situations (2S “... not seriously”). In other words, even taking into account subjective negative experiences, this may point to attempts to accept the existing physical aggression, trying to find positive aspects to justify it because no other alternatives are seen.

The narratives of team sports informants highlighted a tendency to divide into smaller groups, during workouts or competitions, which identify themselves as different by emphasizing negative comments about persons who do not belong to them (1S). Such identification is based on the time of belonging to the team, when “older” team members feel superior to newcomers (2S), or nationality (3S). In the context of group formation, the topic of team/group rallying after workouts and/or competitions emerges, which, if detailed, enables the distinguishing of subjectively perceived subtleties of adaptation. Narratives of 1S and 2S, expanding each other, suggest that team integrity can be temporary and can manifest itself only during the match, as the true negative nature of relationships become evident outside the court, with this being perceived as the team’s weakness. The narrative of 5S shows off-court competition where competition between generations also takes place (“... ready to eat each other ...”). Fitting in with the team is perceived as the problem of the “fitting in” or “not fitting in” person (3S), who by default should himself/herself fit in with the team, but the team remains passive towards the new player. In this context, the “not fitting in” athlete is isolated (avoidance) by demonstrating reluctance to communicate (2S) and/or demonstrating physical distance (4S “... during matches, live in separate rooms ...”).

Another sub-category—degrading of others—shows athletes’ reactions to colleagues’ personal features manifesting themselves by a certain attitude to persons labelled as “conceited”. The fact that as many as five out of seven informants representing different sports confirmed the existence of such problem in their teams or groups indicated prevalence of the phenomenon or “label”. On the one hand, contempt and bullying can be felt from those persons who subjectively see themselves as superior to others (5S), and on the other hand, it is not always possible to precisely distinguish cases referred to as “conceited”, that is, the distinction between demonstrated arrogance, contempt, and reactions provoked by competition. For example, 3S referred to “one of the better players” as “conceited”, who “hindered the microclimate”, without explaining how, but was considering whether it would be better for the team without him; however, he added that he failed to compete with the leader. However, judging from informants’ experiences, the person who subjectively singles himself out and demonstrates certain arrogance receives hostile, malevolent reactions (2S, 4S), becomes isolated, or the person labelled as “conceited” is dealt with off-court (3S “... there were such sparks between them and this was straightaway reflected in the changing room, but this year, we settled it, so ...”).

In the context of competition between athletes, this label is emotionally influential but remains too vague due to subjective perception and therefore is referred to as dangerous because informants’ responses indicate that the decision (e.g., to cope with) can be made by a unified group without the judgment of external arbitrators.

Hostile relationships can turn into intentional harm. The narrative of 7S (wrestling) shows hostility and harm taking a refined, hard-to-spot form; when doing physical exercises, obstacles were formed for the colleague (no help in training is provided) “so that the other fails to do that action ...”. This was, namely, silent sabotage of workout rules, which the victim of hostility subjectively felt but could hardly prove.

Although there were no separate questions about coaches’ behavior, informants named physical and psychological aggression experienced from coaches. This was narrated by representatives of two sport branches: 6S (equestrian sport) and 7S (wrestling). Although coaches’ behavior was perceived as a legacy of the “Soviet” coaching culture (6S) and personal incontinence (7S), this may also reveal certain attributes of coaching culture, wherein physical and psychological aggression is perceived as an instrument to “stimulate” better achievements of athletes, without looking for other forms of motivation. For example, although 3S did not speak about coaches’ behavior, he implicitly expressed his approval of aggressive behavior existing in their interrelationships, interpreting it positively as “hardening”. Moreover, although he himself had become a victim of aggression, he justified its use. Therefore, as far as training culture is concerned, it would be more accurate to use the term sports culture with entrenched positive assessment of physical and psychological aggression. Therefore, the victim of coach’s aggression (7S) shows that athletes can unleash their anger and emotions only in the private space, as the coach–athlete relationship is one-way, while the feedback is cut off, fearing sanctions. This indicates a depressed state with no opportunity to oppose.

The second distinguished category was the nature of bullying in athlete interrelationships and athlete–coach relationships (see Table 3).

On the basis of informants’ answers, the main causes of bullying in the context of sport were distinguished: “jealousy”, “competition”, “appearance (Authors note–body weight)”, “revenge”. Jealousy and competition had commonalities, as hostile behavior was directed against the person’s superior by his/her athletic achievements in order to degrade him/her, harm his/her reputation, hurt him/her and this way affect the dignity, self-confidence, and emotional state of the victim who became the target of attacks, which could affect the match. In this context, the story of 7S is relevant: “...there used to be a lot of bullying, especially mostly because of weight, because anyway, we group by weight category, and the bigger it is, the more it is necessary to bully for being fat, even though you are a Lithuanian champion, still you will stay like this or like that ...”. Duel athletes understand that the fight takes place by weight categories, but stereotypical assessment of appearance prevailing in society is transferred to interpersonal relationships. Such assessments cause psychological pain to the informant, although he/she perceives the cause of bullying only as a reaction to appearance; high achievements mentioned in the story cannot be denied too. Jealousy and competition are permanent in their nature, whereas revenge, on the one hand, is an instantaneous reaction in its nature yet, on the other hand, also exhibits certain lasting approaches because, according to 3S, “older” players take revenge. This suggests that there is a division by age groups, when older athletes subjectively perceive themselves as having “more rights” or privileges than younger ones and claim exceptional conditions for themselves. This way, “taking revenge” can be treated as a tool used to defend privileges subjectively perceived by the members of the “exclusive” group.

Informants’ narratives highlighted verbal and physical aggression. In the case of 3S, physical aggression as material harm occurred when certain items belonging to the athlete were taken away and appropriated, although this happened rarely. In cases of verbal aggression, several aspects can be distinguished—when athletes are criticized for personal failures or blame for the team’s failures is attributed to them. In the second case, criticism used is supplemented by as if additional “weight” in the team’s social context, appealing to the fact that the victim cares about his/her contribution to the team’s success and about the team’s opinion of its particular member. This way, by manipulating the team’s “opinion”, the terrorizing person seeks to increase the subjectively perceived responsibility/guilt of the victim. The narrative of 6S shows how over time, the use of vulgar lexis and jeering suggests the state of inferiority. The victim calls this state “burnout” or the feeling of helplessness after trying to resist by proving his/her value through athletic achievements.

Informants’ responses indicate that physical aggression and threat to use it are employed as a means of impact for both team members and opponents from other teams. As noted above, competition moves beyond the court (public space), where, in some sense, publicly “invisible” negative, that is, aggressive actions can be used. In this context, it is possible to distinguish aggression experienced from both opponents in the team and competitors. Here, the problem of objective and clearly stated assessment reveals itself. For example, narratives of 3S and 4S show that informants realize that objective decisions are impossible in the process of relationships taking place in a closed space without the disinterested attitude from outside, which is not received. This could lead to reconciliation with inevitability of aggressive behavior, seeking no other ways out.

This state of inevitability in the sub-category the athlete’s response to bullying and harassment is outlined by the narrated experiences and considerations of 2S, which revealed hesitation between trying to resist, fight, and reconcile with the situation for fear of further aggressive behavior. This resulted in avoidance (“... I try not to hear ...”) or attempts to justify the experienced feeling of helplessness (e.g., 7S “... I try to avoid all kinds of anger ...”). Such reactions are also promoted by ambient approaches. For example, the narrative of 3S shows that reporting to the team management about experienced violence is perceived as a reprehensible act that is not tolerated by the team; therefore, the victim imparted to persons who were not related to the team in order to avoid revenge, all the more so because retaliatory violence is considered a certain norm, a sign of psychological and physical “strength” (4S, 5S).

Experienced states are revealed by the sub-category the athlete’s emotional state. Informants’ experiences highlighted that violators’ pressure is lasting in its nature, undermining the victim’s self-confidence, and that the victim experiences despair, suffering, and shame, and feels “burnout”, exhaustion, and doubts about his/her as athlete’s career prospects.

On the one hand (sub-category bullying in different age groups), informants’ narratives reveal how reactions to bullying change, whereas on the other hand, show tense relationships between persons of different ages. For example, informant 2S mentioned that over time, certain reconciliation with the aggression experienced takes place, and on the other hand, 3S believed that in adulthood, encounters between conflicting parties become more open and direct. Furthermore, age is perceived as a kind of “status” that “allows” younger people to be treated more rudely compared with peers. Time spent in the team is also perceived as a kind of “status” providing more privileges (sub-category old-timers vs. newcomers). The “blame” for the team’s failures may be shifted onto the younger player, besides discrimination and the manifestation of bullying. On the other hand, the answers reveal that such discriminatory behavior in sport is regarded as a kind of “norm”, a natural part of athletes’ interrelationships. For example, the nature of narratives of 3S and 4S shows that discriminatory behavior or calling bullying a “joke” are not perceived as negative actions degrading the person’s dignity or causing suffering.

Relationships between coaches and athletes (sub-category relationships with the coach) lack openness, for example, when the coach is explaining decisions made, which promotes hostility (2S), interpersonal relationships bear considerable distance (7S). It also becomes clear that the coach can use bullying “instrumentally” as the means of impact on the athlete, seeking “to encourage” him/her (5S and 6S). This demonstrates that, on the one hand, the impact of verbal bullying is perceived, but on the other hand, the narrative of 5S shows the problem of perceiving bullying. In other words, there is a danger that psychological aggression, both used and experienced, may remain unperceived and unidentified. This behavior of coaches in the coach–athlete relationship was generalized and supplemented with the negative impact on the athlete in the narrative of 6S. The problem of the distance between the coach and athlete even more acutely unfolds in the sub-category that we identified as the coach’s position in cases of bullying and harassment. For example, the informant 2S did not dare to tell personal experiences (which can be seen in his narratives discussed above) and conveyed them indirectly by providing the narrative of another athlete who was forced to leave the team. The informant understood that the coach did not know this story because he remained aloof from what was going on outside the court, referred to as “the changing room”. Thus, two territories showed up: one, the public zone encompassing workouts and competitions, where the coach has power, and the other, the non-public zone, where the coach does not interfere, although non-public interpersonal relationships nonetheless influence the way the team functions. Nevertheless, the informant fostered hope that the coach had instruments to solve the problem.

The third category, bipolarity of consequences of bullying in athletes’ interrelationships, revealed what, in the athletes’ opinion, could be treated as consequences of bullying in sport (Table 4).

It is noteworthy that informants’ narratives revealed the consequences of bullying for the individual or social group, but athletes believed that they can have two poles, both negative and positive.

The sub-category “negative impact on the individual’s activity” revealed the motive for personal failures, manifesting itself as the loss of athletic mastery, negative learning outcomes, or drop out of sport. Particularly striking was the reported aspect of drop out of sport, having encountered bullying. However, such a conclusion was made on the basis of observed bullying situations. It should be noted that this problem is encountered by athletes of various sports branches, independent of whether they represent team, individual or duel sports. As stated by athletes, personally experienced bullying, which clearly proves its existence in sport, can lead to poorer academic achievements (4S) or loss of individual athletic mastery. The latter unfolds in the narratives through the prism of psychological bullying directed to sporting activities. In basketball (3S), this can manifest itself by persistent sneering of team members when no points are scored, which seems to program the person’s further failure, performing the same action again—throwing into the basket. Such situation can also occur in other sports, such as equestrian sport (6S). The constant saying of “... you won’t do it anyway” determines failure in sport. It should also be noted that this type of bullying can be demonstrated by both team members and coaches. The latter, according to athletes, exhibits communicative behavior called “casting aspersions” (3S).

Experienced bullying in sport can also cause “negative psychological consequences”. Both continuous physical actions—“pushes” (3S)—and psychological actions—“pressure” (4S)—can form the feeling of lack of self-confidence. This feeling eventually moves to other areas of life, “... that I looked at every new thing in such a way that I would still fail, well, there wouldn’t be anything good still ...” (6S).

Bullying in sport can also determine the person’s disassociation from the group, which becomes the individual’s defensive response to various bullying situations. There is no wish to encounter the bully to whom “antipathy” is felt any longer (5S), because memories of bullying do not disappear. However, when bullying behavior in the team or club repeats, the athlete naturally faces the dilemma, “… maybe it’s enough for me, it’s enough to bully me and maybe it’s time to end that career ...“ (5S), which has nothing to do with the wish to engage in sporting activities.

These consequences also naturally determine “the negative impact on the social group”, stated by athletes, that is, on the team or club and its results, as not only “personal goals” are sought in sport (1S). It is as if it is understood that there should be no “anger”, “arguments”, “tension”, or “miscommunication” in team members’ relationships, but in reality, these actions lead to the game that “... straightaway worsens ...” (3S), whereas athletes experiencing bullying “... do not help the team at all ...” (5S).

However, stating the consequences of bullying, informants do not shy away from taking the observer’s role, and much more willingly remember the persons they saw in the victim’s roles.

However, according to respondents, bullying experienced in sport may not always mean negative things. The behavioral pattern “an eye for an eye” is becoming a norm in athlete relationships, whereas persons who have “withstood” it in sport state that they do not attach much importance to bullying. They choose the strategy of “... just reconciliation ...” (7S), which shows that bullying in their personal relationships remains and is transmitted from one generation to another. Most probably, such patterns of communication among athletes and the coach’s behavior with athletes inevitably leads to situations where bullying will be experienced over and over again. Moreover, because there is a strong need to engage in sporting activities, all that remains is the justification of such behavior. This manifests itself by athletes’ opinion that the ability to withstand bullying, not to break, will build a strong character and harden the athlete, “... I’m sort of glad that I have stayed in such medium, because when you stay in such place (of the victim), you realize that other things that earlier looked very terrible are not so terrible, on the contrary, they have kind of grown such skin to me so that it’s easier for me ...” (2S).

## 4. Discussion

The conducted research primarily confirmed the statement that bullying does not disappear anywhere over time and that the likelihood that victims will experience bullying in further life remains considerable [35]. Although the analysis of the impact of sporting activity and its processes on the person, on development of his/her physical or psychological skills often highlights its positive meaning, negative aspects, including bullying, should not be ignored. However, in this case, research has mostly focused on negative relationships of children or adolescents, also including bullying, or on the analysis of experiences of professional athletes. Meanwhile, there is little interest in the problem of bullying among youth and young adults engaged in amateur sport due to the prevailing approach that bullying victimization decreases with increased age [36] and that one’s personality when reaching the adult age group distinguishes itself by maturity, having already achieved identity with amateur sport [37]. However, athletes’ narratives reveal manifestations of the existing, although partially masked, problem of bullying in organized sport among young adults.

As a result of the analysis of research results, Figure 1 presents the main trends summarizing the study.

This study shows several factors promoting bullying in sport, although the lack of research allowing for the comparison of the research findings with the research conducted on bullying in sport in the adult context has also revealed itself. The outcome of this study allows us to distinguish individual and contextual factors determining bullying in interpersonal relationships. The first level includes individual characteristics of conflict participants, as well as the contextual level, a group of factors encompassing established traditions justifying negative behavior. For example, revenge due to failure can be instantaneous, but jealousy is permanent/long-term in its nature. On the other hand, jealousy provoking unfair competition is one of the significant factors, but it is involved in the interaction with other factors. One of such factors is the person’s perceived status in the team, which is characterized by the athlete’s age. This factor was also found by Kerr et al. [17], however, our study demonstrates the multiplicity of this factor. Over time, that is, as one reaches an older age, perception of bullying may undergo certain adaptation while observing and experiencing bullying, and the nature of reactions to bullying may change also. It was noted that victims are trying to ignore acts of bullying. Due to these processes, bullying may remain unidentified; for example, in the beginning of interviews, many athletes stressed that there was no bullying in their team/group or sports branch. Such attitudes are influenced by traditions existing in sport, where bullying and age discrimination are not accepted as negative, anti-social behavior. In this context, linking of age and time spent in the team with social status may have influence. For example, it was noted by [38] that there are links between bullying and social status perception at school as certain popularity among children or teachers. Our study shows that social status perceived in athletes’ environment is not necessarily associated with the athlete’s achievements (achievements can provoke jealousy manifesting itself through bullying) as much as it is with such criteria as older age and longer time spent in the team. The sub-category of old-timers and newcomers in athletes’ interpersonal relationships, which revealed itself in our study, is a self-evident norm. According to [39], college athletes seem willing to do anything that veteran players demand in order to be part of the team’s “inner circle”. Socialized athletes form closed groups that enjoy broad privileges, defending them and organizing bullying referred to as a “joke” against newcomers and younger athletes. Because younger athletes know these unwritten rules, they are forced to satisfy the demands of older athletes, this way avoiding direct aggression. Therefore, the said tradition can be described as a refined, hidden form of aggression and the cause of open aggression. Victims experience shame, are afraid to resist, avoid even more active actions, and find no support. In this context, the problem of the coach, who directly and indirectly supports the culture of violence, showed up. In the case of our study, it can be seen that, first, there is opposition between athletes and coaches who are identified as representing “Soviet traditions”. That is, coaches use physical and psychological aggression against athletes as a certain instrument to encourage athletes to make greater efforts. Second, coaches keep aloof from solving interpersonal relationship problems between athletes outside the court, which is referred to as “the changing room”. This also confirms the results of previous studies conducted by the authors of this article [19], where coaches were interviewed. Third, a certain distance maintained by coaches, which prevents open athlete-coach relationships in both solving sports issues and trusting interpersonal relationship problems, becomes evident.

Such behavior of coaches can be directly related to the primary universally raised role of the coach—to help athletes to improve their performance [40], which is accompanied by the authority given to the coach and to the lack of competence in the area of interpersonal relationships [41]. All of it supports the bullying-friendly environment in which the athlete experiencing bullying feels helplessness, despair, and, in trying to fight with aggressors, feels defeated and “burned out”.

Yildiz [42] states that high levels of burnout that victims might experience because of bullying may reduce individual performance as well as team performance. Furthermore, higher levels of burnout contribute to attrition (losing a well-performing player prematurely), which could have a significant effect on the team’s overall performance and competitiveness. Our study shows that attrition caused by bullying, the inability to oppose, and experienced shame can lead to thoughts of dropping out of sport.

This study also revealed different consequences of experienced bullying and demonstrated the specificity of their treatment in sport. If research analyzing consequences of bullying in various contexts states—namely, its negative impact on the person’s psyche, activity, or development—reveals potential problems related to substance, alcohol, or illicit drug use; emerging sleep disorders; headaches or dizziness; stomach aches; or back pain [6] in the sports context, bipolarity of consequences of bullying distinguished by athletes shows up. This means that alongside negative consequences, such as worsened academic performance, loss of athletic mastery, drop out of sport, experienced social isolation, and emergence of lack of self-confidence, positive benefit of experienced bullying can arise as per the athletes’ opinions. Surveyed athletes believe that even if manifestations of bullying in sport are present, personal experiences of bullying provide the opportunity to become a stronger, more combative and better athlete. It should be maintained that such a situation can be explained by various socio-cultural mechanisms, and in particular, by the theory of sport as a violent subculture [43] or the theory of social learning [44]. Athletes act in the environment where situations that are considered otherwise intolerable are accepted in sport as common and even necessary to achieve goals. That is, athletes socialize in an environment that often urges them to maintain aggressive tactics in order to win. Furthermore, factors such as respect for authority, control of one’s life, perception of hierarchy, the ability to sacrifice for the game, striving to stand out, or inability to set limits are identified as common and are valued in sport [45]. In other words, modern sport is no longer just a competition, seeking to find out who is the first. It becomes a social medium in which athletes check their identity attached to sport [46].

However, it is likely that eventually such “positive” benefit of experienced bullying will become a threat to the person’s mental and physical health. Primarily, becoming a “stronger” athlete can lead to an increased risk of trauma during sports [45]. Qualitative research, which has analyzed the effects of aggression on victims and revealed such long-term consequences as alcohol and drug use, problems caused by gambling, and low self-esteem, also enables the prediction of long-term consequences of bullying [47].

Also, in order to envisage guidelines for future research, we refer to the position imparted by Fields et al. [48], who stated that “future research must provide accurate national estimates of the incidence of sports-related violence among youth, identify associated risk factors, evaluate preventive interventions and identify effective methods of distributing and implementing evidence-based interventions” (p. 32). According to the authors, the magnitude and distribution of the burden of sports-related violence should be monitored. As for bullying in athletes’ interrelationships, as well as any other form of intolerable behavior, all organized sports participants are obliged to have a common understanding what that bullying is, and there must be a serious attitude to consequences and an appropriate response to situations where bullying occurs. Achievement of this goal necessitates studies that will become the basis for formation of appropriate organizational culture, which actually is “… an important buffer to bullying” [14].

## 5. Conclusions

On the basis of the analysis of the research data, we distinguished three generalized categories: intolerable perception of behavior, the nature of bullying, and bipolarity of consequences. Every category was detailed by sub-categories highlighting the nature, causes, and consequences of bullying, accepted by athletes in the contexts of their emotional state and career. We found that the factors falling into these categories were interrelated and supplementary to each other; therefore, they should be analyzed in a complex way, as bullying is determined not by some single factor but by the totality of them, functioning as a kind of well-established flawed tradition supported by the cultures of the sports organization and the sport.

This study broadens our knowledge of causes of bullying in sport and its specificity, pointing out the problems of perception and identification of bullying, due to which both physical and psychological aggression may remain unrecognized by athletes themselves and their coaches, being perceived as some kind of “natural” behavior inherent to sport. Distinguished categories and sub-categories can serve as a basis for further research, both qualitative and quantitative, going deep into the causes of bullying and investigating its prevalence in different branches of sport. There is also a visible need for research into the relationship between hypothetical school bullying and a possible transference on sports practices with the same athletes. There are, however, limitations of the research and guidelines for further research. This study involved athletes representing various sports branches; therefore, no conclusions attributable to a specific sports branch or group of sports branches can be drawn. In the future, it would make sense to conduct research with subjects involved in specific sports branches or groups of sports branches, choosing either young adults or adolescents. The study involved athletes from only one country; therefore, in the future, the situation of athletes involved in the same sports branch or group of sports branches in several countries on the analyzed issue could be compared.

## Figures and Tables

**Figure 1 ijerph-16-04685-f001:**
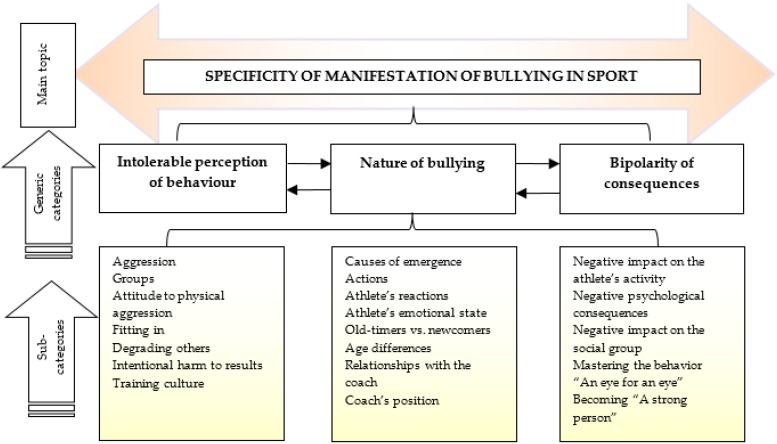
Specificity of bullying in sport: causes, actions, and consequences.

**Table 1 ijerph-16-04685-t001:** Informant sample.

Informant Code	1S	2S	3S	4S	5S	6S	7S
Gender	Female	Male	Male	Male	Female	Female	Female
Age	22	19	20	27	29	28	20
Sports	Rugby	Football	Basketball	Boxing	Athletics	Equestrian sport	Wrestling
Years in sport	9	10	16	12	11	17	9

**Table 2 ijerph-16-04685-t002:** Intolerable perception of behavior in interrelationships between athletes.

Sub-Category	Interview Statements Justifying Distinguished Sub-Categories
Aggression	1S: “... she just started scolding the girl a lot and punched her in the face ...”. 3S: “... there is some other player in my position, so, maybe he plays there like I do, so there are cases that he biases other teammates or spreads some rumors there ... such things existed, really, but ...”. 5S: “... such sneering used to be, and it used to be unpleasant and painful, but if you have that thick skin, you don’t pay attention, but for another person it might be different ...”. <...> “... only psychological, you won’t call it otherwise, because I have never seen violence, someone fighting, but psychological, just ...”.
Attitude to physical aggression	1S: “... we are playing rugby; we are not dancing ballet ...”. 2S: “... once we had a fight ... but not seriously, just it happened so, we had a quarrel and he wanted to sort out and we sorted out ...”. 3S: “... you can see the good side, because things like these harden, and it didn’t happen every day, but did happen ... of course, it used to be hard ...”.
Groups	1S: “... during breaks in the competition or during workouts, all group to bunches and, let’s say, go to drink water or something, but usually there are such groups where several girls interact ... are just trying to disassociate themselves ... gossip about each other ...”. 2S: “... there were such people who thought they shouldn’t associate with newcomers, they felt superior there, better at something ...”. 3S: “... those foreigners are divided into groups, because, anyway, if there are Americans, then Americans communicate with each other, if from the Balkans, they interact with the Balkans, and this way groups are formed and somehow they are treated differently than Lithuanians anyway ...”.
Fitting in	5S: “... the generation was extremely competitive for everything, it seems they grudge everything, all the time everything is for me, if you say something, they are ready to eat each other ...”. 2S: “... they would show that they didn’t like you or that they didn’t want to talk with you or the like ...”. 4S: “... communication is colder, they avoid that person, don’t want to have any contact with him, during matches, live in separate rooms... just, just the distance is kept ...”.
Degrading others	3S: “... he wanted to be a leader, but he didn’t succeed; therefore, he looked at that our leader heavily and there were such sparks between them and this was straightaway reflected in the changing room, but this year, we settled it, so ...”. 4S: “... they put on airs, they get into the ring, you know, and that arrogance finishes ... so those who talk and don’t do are disliked very much, you know ... they are slandered and the like ...” <...> “... try to pretend to be someone that they are not, act like stars, don’t see anybody equal to them, think that they always do the best way...”. 5S: “... you realize that maybe he doesn’t like, maybe he puts on airs because he is good and I am not as good and not worth that communication ...”.
Intentional harm to results	7S: “... there are several girls who don’t get along, but you can’t object here, if they grouped you, then that’s all—you work ... well, then quality suffers, but then the coach, how to say, says more strictly and you have to overcome yourself...”. <...> “... there are actions, for example, some exercises, one should not oppose another and just let do: usually resist so that the other fails to do that action ...”.
Training culture	6S: “... there is a big problem with coaches in Lithuania, because there are still many Soviet school coaches, who grew up in these Soviet times where, in short, the only means of upbringing, how they say, is the birching ...”. 7S: “... if you object to the coach, you’d better get out of here ... if you’re unhappy, you just clench your teeth and work ...”. <...> “... coaches raise, raise, she loses, then they yell at us, you can say ... strictly moralize ... and then we do what, we can’t object to anything, we keep silent, and then when we go back to the rooms, we are outraged, scream, calm down ...”.

**Table 3 ijerph-16-04685-t003:** The nature of bullying in athlete interrelationships and athlete–coach relationships.

Sub-Category	Interview Statements Justifying Distinguished Sub-Categories
Causes of emergence	1S: “... jealousy ...” <...> “... I’m not worse than her, I had to be taken instead of her ...”. 3S: “... older players who, for example, did not get a pass from the player, just in the changing room come and teach by violence ...”. 6S: “... there was jealousy when someone from the team used to get a better horse ... then that’s it, part of the team already turns away from you ... for some month, they turn away from you purely out of jealousy ...”. 7S: “... there used to be a lot of bullying, especially mostly because of weight, because anyway, we group by weight category, and the bigger it is, the more it is necessary to bully for being fat, even though you are a Lithuanian champion, still you will stay like this or like that ...”.
Actions	1S: “... not to speak of physical aggressiveness, just emotionally aggressive—liked to yell, get angry ...”. 2S: “... when our team would lose, it actually would seem that everyone was putting all the blame on you, although you weren’t the most guilty there ...”. 5S: “... even during competition, you are running, you are overtaken, at half-circle there, that disliked group fellow is standing and sneering at you… you failed, stumbled over the hurdle, you are the last and again such: oh, you’re useless here ...”. 3S: “... two players started fighting in the changing room for the same position ... no one else knows these things, neither the media nor anybody, this is just such internal information ...”. 7S: “... girls, if she’s in a bad mood, she’ll show you and batter you, and everything ...”.
Athlete’s reactions	1S: “... I just didn’t say anything ...”. 2S: “... I usually try not to hear such things ... when there were cases when they said something more rudely or so, you still need to stand up for yourself, you say back or show in some other ways ...”. <...> “... you won’t complain anyway, if you complain they will stifle you even more ...”. <...> “... and somehow I weighed and thought: I’ll endure, I’ll keep trying ... You need to stand for yourself ...”. 5S: “... I have a thick skin, I can retort ...”. 7S: “... I keep silent all the time ... I’ll better suffer, let them yell at me, but I’ll not get involved ... I just keep silent and nod ... I’m like a sponge, absorb everything, try to avoid all kinds of anger ...”.
Athlete’s emotional state	3S: “... I’m not saying it’s easy now, still that psychological pressure exists ...”. 5S: “... you start thinking, maybe really I should finish that career, maybe I should not do sports really, maybe I should not disgrace any longer ...”. 6S: “... you sneer at me, but I will still do how I wish and eventually you will anyway be jealous of me, for I will reach what I want ...”. 7S: “... well, you can’t change your partner, you still have to work with the same, so, you just suffer, maybe crying, but you’ll still have to work ...”.
Age differences	3S: “... when we were kids, bullying was more behind your back, there were more gossips, while adults more often just say it straight, then, of course, there may be a physical conflict, but I think that adults, anyway, more often say directly ...”. 5S: “... adults are ruder, this hurts more, you already take things more personally and your mind works differently ...”. 5S: “... the workout takes place, the younger player understood the combination wrongly, so instead of saying nicely, yell at him with swear words ...”.
Old-timers vs. Newcomers	2S: “... those players who have played for a longer time would shift the blame on younger ones, these newcomers, therefore, you would feel as if like alone ...”. 3S: “... there are tasks that have to be done by every young athlete doing sports for the first year, but this is such well-established rule and everyone knows it, but not going as far as bullying or something, or sport, because these are such tasks that that youngster must perform. Let’s say the team goes on a trip by bus, he can’t sit where he wants: he must let the older one choose seats and only then the young player can have a seat ...”. 4S: “... it happens that newcomers who start to attend in other cities; so, they do something to them when they are asleep when they fall asleep ... play a joke, for example, put toothpaste on the eyes, you know, or the like or pour down water, so, there have been such things ...“.
Relationships with the coach	5S: “... I don’t know if it’s bullying here, but coaches always stressed to us that we were fat, that we had to lose weight, we, fat, would not be able to run...”. 6S: “... my career was developing starting with very good coaches who knew how to communicate, to professionals who can’t train at all, in that sense that they don’t know how psychology works and how the coach’s psychology should work ...”. 7S: “... for example, the team coach, so, I don’t even talk with him about personal things, I couldn’t even talk with him ... we have a personal psychologist, so, we talk with him ...”.

**Table 4 ijerph-16-04685-t004:** Bipolarity of consequences of bullying in athletes’ interrelationships.

Sub-Category	Interview Statements Justifying Distinguished Sub-Categories
Negative impact on the athlete’s activities	4S: “... for a long time, my academic achievements suffered, so, for that reason it would seem that now I would do something differently, you know. So, ... maybe that achievements suffered ...”. 2S: “... I say: that bullying entirely depends on a person. It can push you in a positive direction or negative. Of course, the vast majority was really influenced negatively, dropped out of sport ...”. 3S: “Sometimes there are cases when it’s not funny, say, they are laughing at you as the basketball player, that you miss the three-point, everyone there: you miss, you miss and then something happens in your head, like, I don’t know, like such disturbance, you throw during the match and you are afraid that if you don’t score, they will keep saying that you don’t score. Therefore, you throw being concentrated, psychologically you think not to miss, which really doesn’t help but only hinders. Then you miss and get even more nervous, then you stop throwing, although you have to throw. So, such are cases, I had experienced myself ...”. 6S: “Continuous repetition that, anyway, you won’t do it and knowing that most often I eventually fail. Let’s say, I know my mistakes myself, but when the coach comes and casts aspersions on you, I would already know, approximately knew before the start that it would be bad anyway ...”.
Negative psychological consequences	2S: “Obviously, that bullying had its own such ... maybe it affected self-confidence more. The biggest problem was with self-confidence ...”. 3S: “Maybe there are psychological, I don’t know physical. As to me, maybe lack of self-confidence, because anyway, I was the weakest, when they used to push me or something, when I wouldn’t be able to do anything, I would feel not that much inferior but such distrust might have appeared a little bit, I don’t know ...”. 6S: “It seemed as if the failure was kind of fated, that I looked at every new thing in such a way that I would still fail, well, there wouldn’t be anything good still ... that view: it’ll be bad anyway, it’ll be bad anyway ...”. 4S: “... most probably would become depressed, avoid those people who ...”.
Negative impact on the social group	1S: “… but maybe she is a very good player and she is really badly needed on the court, in the team. And maybe our own as athletes’, as team’s composition would suffer. Because maybe without that good athlete we will really find it difficult to participate in matches ...”. 2S: “… there can’t be any anger, results suffer because of that, that’s it. … well, anyway, everyone is seeking one result and personal goals ...”. 3S: “… yes, yes, it manifests itself and the game gets worse right away. Obviously, because when there is such negative tension, straightaway there appears miscommunication, anger, arguing, it is immediately reflected in the play ...”. 5S: “… and there are cases that those players don’t help the team at all. They defend poorly, they are not concentrated, they both feel tension between them. When the team has to play and win like one fist, but this doesn’t happen, the whole team suffers because due to those two players no result is achieved ...”.
Mastering the behavior “an eye for an eye”	1S: “... another person behaves the way you behave with him ...”. 4S: “… if someone starts beating me? I would definitely strike back. So all, I think, all would strike back there, you know. I don’t really think that someone would keep sitting there. ... I personally would definitely kick back ...”. 5S: “… if I’m treated well, I treat well too. If badly, it’ll be bad for you too ...”.
Becoming “a strong person”	2S: “I’m sort of glad that I have stayed in such medium, because when you stay in such place [*of the victim*], you realize that other things that earlier looked very terrible are not so terrible, on the contrary, they have kind of grown such skin to me so that it’s easier for me ...”. 3S: “I was worrying that I was weaker, smaller. Now that’s exactly what I’m trying to exploit … but now it’s really not so that I would think I’m weaker and can’t take full advantage of all opportunities ...”. 4S: “… and if there were some insults, those bad emotions, I don’t give prominence to them. So I really ... gave a lot in life and prepared for that life school, so really such thing hasn’t remained in me ...”. 6S: “I didn’t feel worse than those who sneered at me; on the contrary, I felt better than them ...”. 7S: “Now, I look at these, how shall I put it, disagreements completely differently. Well, I wouldn’t react at all, I’d just laugh or start asking why me. I would get into an argument, which I wouldn’t do before, I would immediately start crying and run away, and now, I say what you want ...”.

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
