# Peer review of "Encounter with Bullying in Sport and Its Consequences for Youth: Amateur Athletes’ Approach"

_ijerph, 2019, doi:10.3390/ijerph16234685_

Round 1

Reviewer 1 Report

Title: Encounter With Bullying In Sport And Its Consequences For Youth: Amateur Athletes´Approach

Abstract:

- Numbers from zero to ten must be written here and full text (e.g. line 17, …involved 7 organized sports…) except when the number is used with a letter to specify an athlete (e.g. 6S…).

- Authors do not specify which computerized software have used to perform the content analysis as kulatunga, Amaratunga and Haigh (2007) recommended in the reference nº30. In addition, they do not specify the software used in the section 2. Materials and Methods.

Introduction

- In the line 64, authors should be written because it is a little bit confusing. The reader could not know if those authors are from reference nº 10, nº4 (before) or nº8 (after).

- It is only an advice, in the line 75, there are nine references and maybe could be enough with three-five (…in the recent decade [11-20], which is to be…).

- It would be interesting to add from where is coming the bullying in sport in the lines 85-93 (e.g. from others team mate, coach, etc.).

Materials and Methods

- Qualitative research is pretty good explained by authors. Moreover, they add science-based literature and explanations.

- Authors should provide information about how they contacted with the athletes in the lines 128-131 (by some organization, athletes club, association, etc.).

- Did the authors use a voice recorder for the semi-structured face-to-faced interviews (line 133)? Please, specify it.

Results

- The line 179 and 182 should coincide with the same category name: “intolerable perception of behaviour” or “Intolerable behaviour”.

Discussion

- Figure 1 is clarifying.

- Well argued.

Conclusions

- Further research should study the relationship between hypothetical bullying in school and a possible transference on sport practice with the same athletes.

Author Response

Dear reviewer,

Thank you very much for your comments and thoughtful suggestions. Based on these comments and suggestions, we have made minor corrections to the original manuscript.

Abstract.

We corrected numbers of respondents by writing seven instead of 7. Also it is explained in Materials and Methods section (line before Table 1).

We do not use any computerized software in performing content analysis as Kulatunga, Amaratunga and Haigh (2007) emphasizes in their article that ”… computer aided software needs to be used with caution”, because “the use of such software can make the researcher mechanistic and damage creativity”  and can make process mechanistic.

Introduction

- In the line 64, authors should be written because it is a little bit confusing. The reader could not know if those authors are from reference nº 10, nº4 (before) or nº8 (after).

We have corrected this part.

- It is only an advice, in the line 75, there are nine references and maybe could be enough with three-five (…in the recent decade [11-20], which is to be…).

We think we mentioned main research here

- It would be interesting to add from where is coming the bullying in sport in the lines 85-93 (e.g. from others team mate, coach, etc.).

We have corrected – peer bullying

Materials and Methods

- Qualitative research is pretty good explained by authors. Moreover, they add science-based literature and explanations.

- Authors should provide information about how they contacted with the athletes in the lines 128-131 (by some organization, athletes club, association, etc.).

We have provided information about it (now lines 131-138)

- Did the authors use a voice recorder for the semi-structured face-to-faced interviews (line 133)? Please, specify it.

We have corrected – It was used Dictaphone

Results

- The line 179 and 182 should coincide with the same category name: “intolerable perception of behaviour” or “Intolerable behaviour”.

  We have corrected

Conclusions

- Further research should study the relationship between hypothetical bullying in school and a possible transference on sport practice with the same athletes.

Thank You for this advice. We have included this in our conclusion.

Reviewer 2 Report

In the introduction I recommend not to abuse the literal phrases. It is better to write with your own words the ideas of the other authors.

I consider that first paragraph of the method is not relevant.

Explain why the sample is consider as youth if they are from 19 to 28. This range of age is consider as "adult" and the buling phenomenon occurs mainly in adolescence.

Author Response

Dear reviewer,

Thank you very much for your comments and thoughtful suggestions. Based on these comments and suggestions, we have made minor corrections to the original manuscript.

In the introduction I recommend not to abuse the literal phrases. It is better to write with your own words the ideas of the other authors.

Some of them (line 65-66 and 68-69)  we write with our own words.

I consider that first paragraph of the method is not relevant.

We partly agree with this comment. However, we believe that this section reveals the authors' immersion into the philosophical approaches of qualitative research.

Explain why the sample is considering as youth if they are from 19 to 28. This range of age is considering as "adult" and the bulling phenomenon occurs mainly in adolescence.

In this study, by naming the sample, we are guided by Lithuanian youth policy https://socmin.lrv.lt/en/activities/family-and-children/youth-policy. It states that “Youth” is young people aged 14-29.

Also we have made minor corrections in describing research methods.